# Transfer Learning Improving Predictive Mortality Models for Patients in End-Stage Renal Disease

Edwar Macias [1,*], Jose Lopez Vicario [1], Javier Serrano [1], Jose Ibeas [2] and Antoni Morell [1]

1   Wireless Information Networking (WIN) Group, Universitat Autònoma de Barcelona (UAB),
    08193 Bellaterra, Spain; jose.vicario@uab.cat (J.L.V.); javier.serrano@uab.cat (J.S.);
    antoni.morell@uab.cat (A.M.)
2   Nephrology Department, Parc Taulí Hospital Universitari, Institut de Investigació i Innovació Parc Taulí I3PT,
    Universitat Autònoma de Barcelona (UAB), 08208 Sabadell, Spain; jibeas@telefonica.net
*   Correspondence: edwarmatoro@gmail.com

**Abstract:** Deep learning is becoming a fundamental piece in the paradigm shift from evidence-based to data-based medicine. However, its learning capacity is rarely exploited when working with small data sets. Through transfer learning (TL), information from a source domain is transferred to a target one to enhance a learning task in such domain. The proposed TL mechanisms are based on sample and feature space augmentation. Thus, deep autoencoders extract complex representations for the data in the TL approach. Their latent representations, the so-called codes, are handled to transfer information among domains. The transfer of samples is carried out by computing a latent space mapping matrix that links codes from both domains for later reconstruction. The feature space augmentation is based on the computation of the average of the most similar codes from one domain. Such an average augments the features in a target domain. The proposed framework is evaluated in the prediction of mortality in patients in end-stage renal disease, transferring information related to the mortality of patients with acute kidney injury from the massive database MIMIC-III. Compared to other TL mechanisms, the proposed approach improves 6–11% in previous mortality predictive models. The integration of TL approaches into learning tasks in pathologies with data volume issues could encourage the use of data-based medicine in a clinical setting.

**Keywords:** transfer learning; deep learning; mortality prediction





## 1. Introduction

In the era of Big Data, deep learning (DL) is becoming a fundamental piece in the paradigm shift from evidence-based medicine to data-based medicine [1]. The increased availability of information, storage and processing capacity, and DL's capability to exploit complex relationships has allowed DL to significantly impact medical applications supported by Big Data [2]. Although the adoption of technologies that enable the collection of high volume of data in a clinical setting is growing, most medical centers do not have the infrastructure or the volume of patients to benefit from the learning capacity of DL [3]. Thus, integrating information from multiple health centers could significantly improve learning tasks in pathologies usually supported by a small volume of data. Implementing strategies for transferring data among domains could trigger DL solutions in a clinical setting and bring us closer to adopting data-based analysis for supporting clinical decisions.

The process of adapting and transferring knowledge among domains is known as transfer learning (TL) [4]. The interest in TL in the medical field is increasing. In electronic health records (EHR), such as clinical images and biosignals, DL integration in a TL environment is proving to be an option that provides remarkable benefits [5–8]. Due to the capacity to exploit complex relationships that data may have and the data structures in such applications, e.g., spatial dependencies or time series, specialized artificial neural networks (ANN) are commonly used. In such applications, the common TL approach pre-trains an

ANN with data from one domain. Then the learned parameters are extracted for later use in applications in other domains [9–11]. However, this kind of approach is not suitable for data that contain heterogeneous structures because the procedure mentioned above constrains the input to be similar across domains. That is the case for applications that use other types of EHRs, such as those that collect medical measurements of patients in a tabular way. Although there are solutions that incorporate such data into TL approaches [12,13], they use statistical analysis, which does not exploit complex relationships that the data may have. Thus, alternatives that include DL in TL solutions in pathologies with the type of data mentioned above are still an open issue.

Transferring knowledge from high volume data sources to small datasets would allow DL to enhance learning tasks and be used to address class imbalance issues. This effect occurs because of the sudden changes in the patient's health condition. The volume of information generated for such events is smaller than the one associated with the rest of the follow-up. This effect commonly occurs in pathology prediction [14], rare event detection [15], or mortality prediction [16]. In previous work for mortality prediction for patients in end-stage renal disease (ESRD) [17], data imbalance was evidenced. There was a data imbalance in the range of 76 to 94%. Those issues cause low generalization of the learning models on the imbalanced samples, resulting in models whose performance is not acceptable for incorporation into clinical practice.

This work proposes a TL framework that uses information from a massive data source for supporting tasks in pathologies with a small data volume. The framework consists of two TL mechanisms used for sample and feature space augmentation in a target domain. Autoencoders (AE) are used to link both domains as a knowledge extraction mechanism. From AEs, latent representations of data, the so-called codes, are used as information bridges. For the sample increasing mechanism, they are used to create a feature mapping matrix used to transfer samples for a source domain to the target one. For the feature space augmentation, the TL mechanism is based on the computation of the average of the most similar codes of the target with the ones generated in the source domain. This TL framework is evaluated for the improvement of mortality predictive models in patients in ESRD. Volume and data imbalance issues are tackled with information extracted from patients with acute kidney injury (AKI) from the massive database medical information mart for intensive care III (MIMIC-III) [18]. According to our knowledge, this is the first solution that integrates ANNs into a TL framework for solving learning tasks for kidney diseases.

The main contributions of this work are as follows:

- Explore the benefits of using a DL approach to TL in the clinical setting;
- Improve predictive models of mortality in ESRD patients by incorporating knowledge from a more extensive data set;
- Tackle the class imbalance issue through a solution based on TL.

The rest of the manuscript is structured as follows: Section 2 shows all the necessary components for the proposed TL framework Section 3 shows the performance of several experiments, and Section 4 presents the discussion, remarks on such results, and the conclusion of this work.

## 2. Materials and Methods

This section contains the necessary components to support the proposed TL framework. Classic AEs are the backbone of the knowledge extraction in the proposed framework. Moreover, two extensions of AEs widely used in the TL environment are also addressed because they are used for performance comparison with the proposed method. Then, a method that has inspired part of the proposed framework is briefly explained. Finally, the problem that the methods can address is formally defined. Next, the necessary components to understand the proposed TL framework are described.

### 2.1. Autoencoders

An AE is a type of ANN with a mirrored structure that replicates input data **x** to the output of the network **x'** with a minimum error. This mechanism allows extracting complex relationships that the data may have into the network's hidden layers. The deeper the ANN, the more complex the representations are, and usually, more data are needed for avoiding overfitting issues. Figure 1 shows a basic and a deep AE. An AE can be divided structurally into two components: encoder and decoder. The encoder acts as a mapping function $f_\theta$ that transforms the input **x** into higher representation level **h**. Such representation is commonly referred to as a code. The code is mapped back to reconstruction using another mapping function $g_{\theta'}$ called decoder.

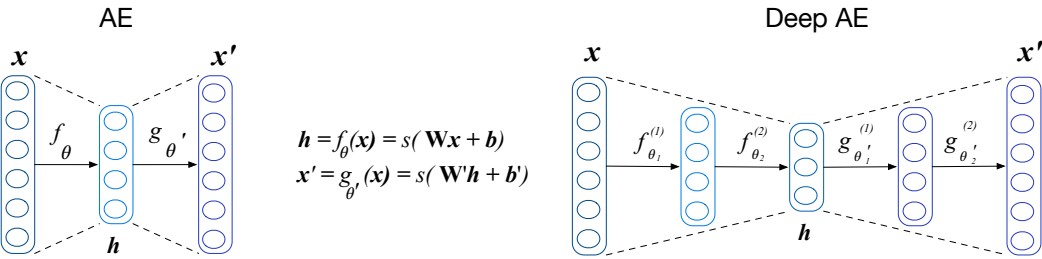

**Figure 1.** Structure of single- and multi-layer AE.

The training of an AE is an iterative process whose purpose is to find proper parameters for the network that minimize the error between the input and its reconstruction. The parameters that are trained are $\theta = \{\mathbf{W}, \mathbf{b}\}$ and $\theta' = \{\mathbf{W'}, \mathbf{b'}\}$, where **W** refers to the weights of the network and **b** their bias. To find the minimum error, the input of the AE is forward propagated through the network. Each unit in the network combines the outputs of the previous layer linearly, and its output is modified by a non-linear function (*s*). This non-linearity allows deep architectures to extract complex representations from massive data. Once the propagations reach the output layer, a cost function $\mathcal{L}$ is computed, and the weights of the AE are updated with the gradient of the error through the network following the back-propagation algorithm [19]. In this work, mean squared error is used as a lost function:

$$\mathcal{L} = \frac{1}{N} \sum_{i=1}^{N} \left\| \mathbf{x}_i - \mathbf{x}'_i \right\|^2, \tag{1}$$

where $\mathbf{x}_i$ represents a sample *i*, and *N* is the total samples in a dataset.

Other alternatives that have shown outstanding performance using AEs in a TL environment are based on the application of stacked denoising AEs (SDA) [20] and its extension marginalized SDA (mSDA) [21]. For the SDA, denoising AEs (DA) are trained. This type of AEs minimize the error between the input and a corrupted version, hence its name. To create the stack of DA, *n* DAs are trained. The first DA is trained with the corrupted version of the input, the second DA takes as input the code of the previous DA, and so on, as is shown in the left side of Figure 2. The training of each level follows the same process as a normal AE. At the end of the *n* trainings, the respective codes are used to create the final stacking that is shown in the right side of Figure 2.

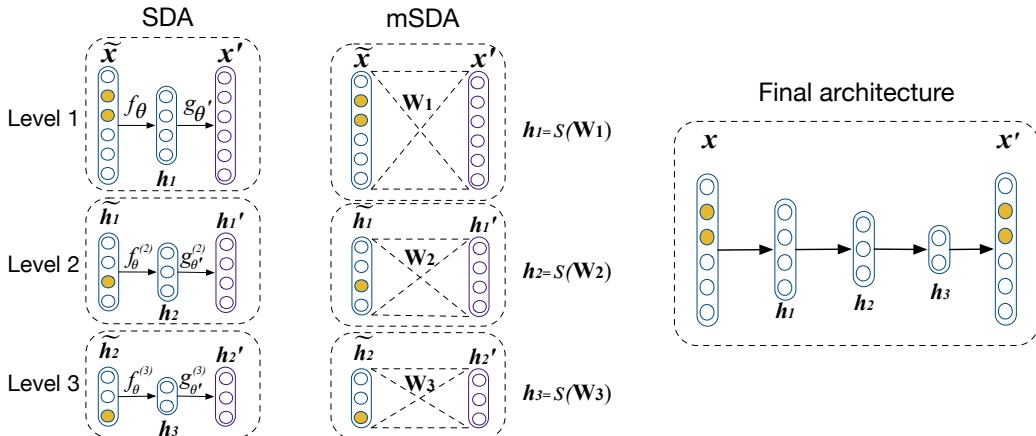

**Figure 2.** SDA, mSDA, and how their latent representations are used to create the final stack.

On the other hand, the term marginalized in mSDA refers to the addition of noise to the inputs $\mathbf{x_i}$ in the iterations of the training process, e.g., different examples may be corrupted in every iteration. Thus, taking this into account, the cost function is transformed to:

$$\mathcal{L} = \frac{1}{NM} \sum_{j=1}^{M} \sum_{i=1}^{N} \left\| \mathbf{x}_i - \mathbf{x}'_{i,j} \right\|^2, \tag{2}$$

where $\mathbf{x}'_{i,j}$ represents the $j$th corrupted version of $\mathbf{x}_i$.

Then, with $\mathbf{X} = [\mathbf{x_1}, \ldots, \mathbf{x_n}] \in \mathbb{R}^{dxn}$, its $m$-times repeated versions $\overline{\mathbf{X}} = [\mathbf{X}, \ldots, \mathbf{X}]$, and its corrupted version $\widetilde{\mathbf{X}}$, Equation (2) is reduced as:

$$\mathcal{L} = \text{tr}\left[ \left( \overline{\mathbf{X}} - \mathbf{W}\widetilde{\mathbf{X}} \right)^\top \left( \overline{\mathbf{X}} - \mathbf{W}\widetilde{\mathbf{X}} \right) \right], \tag{3}$$

and its minimization solution can be expressed as:

$$\mathbf{W} = \mathbf{PQ^{-1}} \text{ with } \mathbf{Q} = \widetilde{\mathbf{X}}\widetilde{\mathbf{X}}^\top \text{ and } \mathbf{P} = \overline{\mathbf{X}}\widetilde{\mathbf{X}}. \tag{4}$$

With a large $m$, i.e., $m \to \infty$, the bias estimation is reduced but the computational cost increases. To mitigate this issue, mSDA includes corruption probability $p$ to a vector probability $\mathbf{q} = [1 - p, \ldots 1 - p, 1] \in \mathbb{R}^{d+1}$. $\mathbf{q}_i$ represents the probability of a feature $i$ surviving the corruption. Thus, the expectation for Equation (4) can be computed and $\mathbf{W}$ can be expressed as follows:

$$\mathbf{W} = \mathbb{E}[\mathbf{P}] \, \mathbb{E}[\mathbf{Q}]^{-1} \text{ with } \mathbb{E}[\mathbf{P}]_{i,j} = \mathbf{S}_{i,j}\mathbf{q}_j, \ \mathbf{S} = \mathbf{XX}^\top \text{ and,} \tag{5}$$

$$\mathbb{E}[\mathbf{Q}]_{i,j} = \begin{cases} \mathbf{S}_{i,j}\mathbf{q}_j\mathbf{q}_j & if \, i \neq j \\ \mathbf{S}_{i,j}\mathbf{q}_i & otherwise. \end{cases} \tag{6}$$

With $\mathbf{W}$, nonlinear function $s$ is applied, then nonlinear features can be extracted as $\mathbf{h} = s(\mathbf{Wx})$. Such nonlinear functions may include tangent hyperbolic (tanh), sigmoid, or Rectified Linear Unit (ReLU).

### 2.2. Hybrid Heterogeneous Transfer Learning

The so-called Hybrid Heterogeneous Transfer Learning (HHTL) proposed in [22] is a TL framework for transferring knowledge between two heterogeneous domains using mSDAs. HHTL solves a learning task related to labelling samples from one domain using information from the other one. The target domain is defined as $\mathbf{D}_T = \left\{ \left( \mathbf{x}_{T_i}, \mathbf{y}_{T_i} \right) \right\}_{i=1}^{n_2}$, and the source domain as $\mathbf{D}_S = \left\{ \mathbf{x}_{S_i} \right\}_{i=1}^{n_1}$, where $\mathbf{x}_{S_i} \in \mathbb{R}^{d_S x 1}$ and $\mathbf{x}_{T_i} \in \mathbb{R}^{d_T x 1}$ are the data and $\mathbf{y}_{T_i}$ labels; $n_1$ and $n_2$ are the totals of the samples, and $d_S$ and $d_T$ their features. The

information to be transferred is the hidden representations extracted from mSDAs for each domain. mSDAs are trained in both domains with k ($k = 1, \ldots, K$) hidden layers, as is illustrated in Figure 3. Then, latent representations $\mathbf{H}_{S,1}, \ldots, \mathbf{H}_{S,k}$ and $\mathbf{H}_{T,1}, \ldots, \mathbf{H}_{T,k}$ are extracted and then related through mapping matrices, $\mathbf{G}_k$, as is shown in Figure 3. These matrices acts as TL bridges for the hidden representations in both domains. To find each $\mathbf{G}_k$, they minimize the objective:

$$\min_{\mathbf{G}_k} \left\| \mathbf{H}_{S,k} - \mathbf{G}_k \mathbf{H}_{T,k} \right\|^2 + \lambda \left\| \mathbf{G}_k \right\|^2. \tag{7}$$

Once $\mathbf{G}_k$ is computed, new samples $\mathbf{X}_S^*$ along with its hidden representations $\mathbf{H}_{S,k}^*$ can be transferred to $\mathbf{D}_T$, i.e., $\mathbf{H}_{S \to T,k}^* = \mathbf{G}_k \mathbf{H}_{S,k}^*$. $S \to T$ refers to the transfer from $\mathbf{D}_S$ to $\mathbf{D}_T$. Then, to solve the learning task, they create a new feature space with the hidden representations of $\mathbf{D}_T$, i.e., $\mathbf{Z}_T = \left[ \mathbf{H}_{T,1}^\top \ldots \mathbf{H}_{T,k}^\top \right]^\top$. Then, a classifier $\{(\mathbf{Z}_T, \mathbf{y}_T)\}$ is trained. With the latent transferred representations, a similar feature space $\mathbf{Z}_{S \to T} = \left[ \left( \mathbf{H}_{S \to T,1}^* \right)^\top \ldots \left( \mathbf{H}_{S \to T,k}^* \right)^\top \right]^\top$ is created. Finally, with the trained classifier, they predict over $\mathbf{Z}_{S \to T}$ the labels for $\mathbf{D}_S$ samples.

Part of the sample augmentation for the proposed approach is based on the computation of $\mathbf{G}_k$, with the difference that we only use it to relate the codes of the AEs and not the rest of the latent representations of each hidden layer. Hence, we compute a single $\mathbf{G}$.

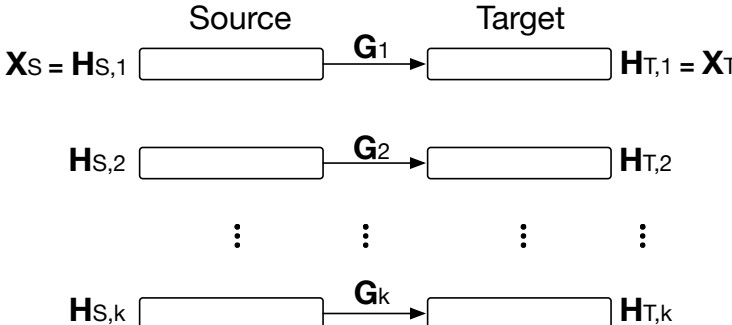

**Figure 3.** HHTL for transferring hidden representations, **H**, between source and target domain. **H** are extracted from trained mSDAs.

### 2.3. Problem Definition

Given a set of labeled data from the source and target domains, $\mathbf{D}_S = \left\{ \left( \mathbf{x}_{S_i}, \mathbf{y}_{S_i} \right) \right\}_{i=1}^{n_1}$ and $\mathbf{D}_T = \left\{ \left( \mathbf{x}_{T_i}, \mathbf{y}_{T_i} \right) \right\}_{i=1}^{n_2}$, respectively, where $\mathbf{x}_{S_i} \in \mathbb{R}^{d_S x 1}$ and $\mathbf{x}_{T_i} \in \mathbb{R}^{d_T x 1}$ are the data and $\mathbf{y}_{S_i}$ and $\mathbf{y}_{T_i}$ their labels; $n_1$ and $n_2$ are the total of samples, and $d_S$ and $d_T$ are their features. The aim of TL in this work is to improve the learning task in $\mathbf{D}_T$ with information from $\mathbf{D}_S$. The transfer of knowledge is carried out by managing codes of trained AEs from both domains in two manners. The first one follows the next steps:

- Transfer samples from one domain to another through the computation of a feature mapping matrix **G**, as in HHTL.
- Map codes from one domain to the other one using **G**.
- Transfer a sample $\mathbf{x}_S^*$ to $\mathbf{D}_T$ through $\mathbf{G}\mathbf{h}_S^*$, where $\mathbf{h}_S^*$ is the code of $\mathbf{x}_S^*$.

The second mechanism attempts to increase the feature space of $\mathbf{D}_T$ with the average of the most similar codes, computed by a similarity metric, the Euclidean distance between the codes, that compares each code from $\mathbf{D}_T$ with the entire set of codes from $\mathbf{D}_S$. The increase in samples and features may reinforce the learning task in $\mathbf{D}_T$. Figure 4 shows a scheme of the mechanisms that are used to enhance the learning task in $\mathbf{D}_T$.

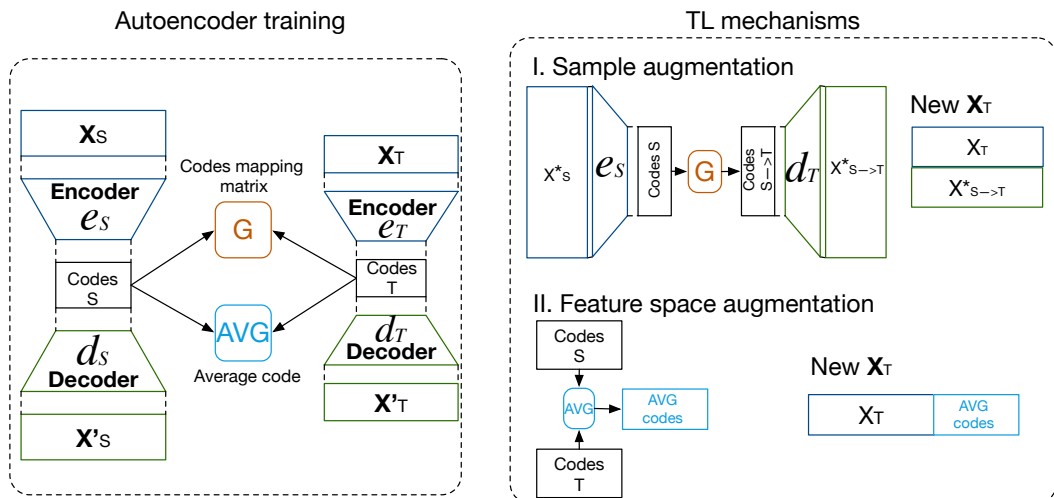

**Figure 4.** Scheme of proposed method for transfer of samples between domains and feature space augmentation to support learning tasks in the target domain. In the training stage, codes are extracted, and **G** and average codes (AVG codes) are computed as TL bridges between both domains.

*2.4. Proposed Method*

The proposed approach is motivated by the availability of massive sources of medical data and the potential benefits of integrating them to encourage the adoption of data-based medicine. This integration makes it possible to exploit the learning capacity that DL has on massive data. Thus, two TL mechanisms were proposed to improve performance in learning tasks in a clinical setting. Specifically, the predictive ability of mortality predictors for patients in ESRD was evaluated using a TL framework. It was proposed to apply TL approaches to augment both samples and feature space in $mathbf{D}_T$ using information from $mathbf{D}_S$. As mentioned above, these mechanisms can address class imbalance problems to improve the predictive ability of previous work for mortality models in ESRD.

In the proposed framework, both domains contained labeled samples. AEs were used to extract data representations into their codes for sample and feature space augmentation. Thus, the framework relied on two main components:

- Sample augmentation using a mapping matrix **G**, encoder and decoder functions in both domains to transfer and reconstruct codes from $\mathbf{D}_S$ in $\mathbf{D}_T$;
- Feature space augmentation based on the computation of the average of the most similar codes.

2.4.1. Sample Augmentation—TLCO

For augmenting samples in $\mathbf{D}_T$, a three-stage TL mechanism was used. Initially, from both domains, AEs were trained, and the codes were extracted to compute a mapping matrix **G**, as in HHTL. It is worth mentioning that, unlike HHTL, in our approach, we reinforce knowledge transfer by considering the reconstruction of the codes of one domain using the decoders of the other domain. We refer to this method as TL by codes or TLCO. In a second stage, **G** is used to transfer codes from $\mathbf{D}_S$. Thus, $\mathbf{H}_S^*$, produced by data $\mathbf{X}_S^*$ in $\mathbf{D}_S$ were first transferred to $\mathbf{D}_T$. Then, the decoder function in $\mathbf{D}_T$ reconstructed the transferred codes in such a domain. The parameters of the decoder function of trained AEs in each domain allowed the reconstruction of their codes. The decoders in the opposite domains and the mapping matrix between the codes can be used as a reinforcement mechanism for cross-domain knowledge transfer. Once the samples were reconstructed, they were used to increase $\mathbf{D}_T$. This last step tackled the class-imbalance issue. Figure 5 illustrates how this TL mechanism was carried out using datasets from kidney diseases. Detailed steps of the proposed method are also provided in Algorithm 1.

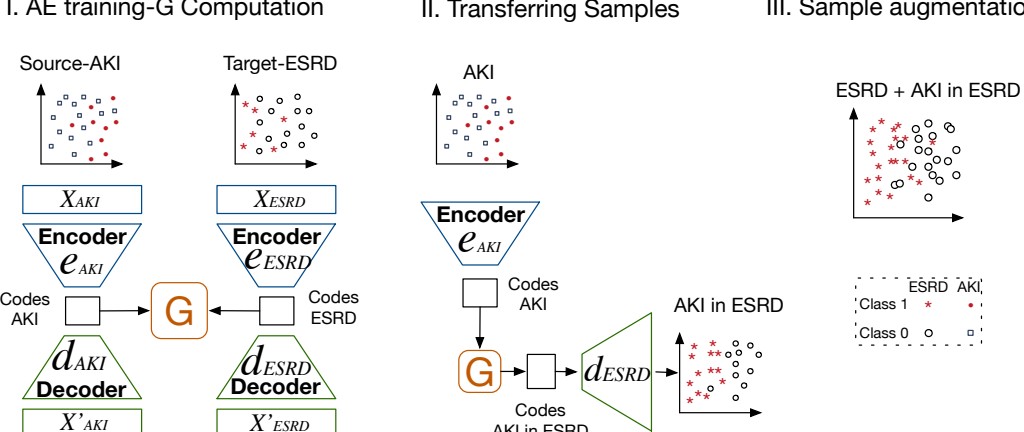

**Figure 5.** Scheme of proposed method for transfer of samples between domains and the support of a learning task in the target domain using TLCO.

---

**Algorithm 1:** Increasing samples using TLCO

---

**Input:** Data from both domains, $\lambda = 0.001$: $\mathbf{D}_S = \left\{ \left( \mathbf{x}_{S_i}, y_{S_i} \right) \right\}_{i=1}^{n_1}$,
$\qquad \mathbf{D}_T = \left\{ \left( \mathbf{x}_{T_i}, y_{T_i} \right) \right\}_{i=1}^{n_2}$

1 Train AEs with $\mathbf{X}_S$ and $\mathbf{X}_T$. Extract encoder ($e$) and decoder ($d$) functions from both domains, and the latent representations-codes $\mathbf{Z}$:

$\mathbf{Z}_S = e_S(\mathbf{X}_S), \mathbf{X}'_S = d_S(\mathbf{Z}_S)$
$\mathbf{Z}_T = e_T(\mathbf{X}_T), \mathbf{X}'_T = d_T(\mathbf{Z}_T)$;

2 Learn heterogeneous feature mapping $\mathbf{G}$:

$$\min_{\mathbf{G}} \|\mathbf{Z}_S - \mathbf{G}\mathbf{Z}_T\|^2 + \lambda\|\mathbf{G}\|^2;$$

3 Augment samples in $\mathbf{D}_T$ with samples from $\mathbf{D}_S$:

$\mathbf{X}^*_{S \to T} = \mathbf{G}^\top \mathbf{X}^*_S$
$\mathbf{X}^*_T = \begin{bmatrix} \mathbf{X}_T & \mathbf{X}^*_{S \to T} \end{bmatrix}, \mathbf{y}^*_T = \begin{bmatrix} \mathbf{y}_T & \mathbf{y}_S \end{bmatrix}$

Note: $S \to T$ refers to the transfer from $\mathbf{D}_S$ to $\mathbf{D}_T$.

4 Train a classifier $f$ with $\left\{ (\mathbf{X}^*_T, \mathbf{y}^*_T) \right\}$

**Output:** Classifier $f$

---

2.4.2. Feature Space Augmentation—TLAV

For feature space augmentation, the TL mechanism was based on the computation of averaging the most similar codes from $\mathbf{D}_S$ to codes in $\mathbf{D}_T$. We refer to these as the average codes or $AVG_{codes}$. They increase features for every sample in $\mathbf{D}_T$. We refer to this approach as TL by $AVG_{codes}$ or TLAV. As the information that best represents the data after AEs' training is encapsulated in their codes, this approach used the $AVG_{codes}$ as extra features that may enhance the predictive capacity of learning models.

The proposed method is summarized into three stages (see Figure 6). Initially, AEs were trained in both domains, and their codes were compared. For TLAV, it is hypothesized that similar codes represent similar information even from different domains. Thus, every code from $\mathbf{D}_T$ was compared with all the codes from $\mathbf{D}_S$. The Euclidean distance was computed as a similarity metric for the comparison (see Equation (8)). Then, the most similar codes were filtered based on a similarity threshold, $\epsilon$, which indicates the percentage of the most similar codes. Based on $\epsilon$, sets of $n_3$ (as in Figure 6) codes from $\mathbf{D}_S$ were

extracted for each code in $\mathbf{D}_T$. Then, in the second stage, the codes' sets were summarized in their average to find a more robust representation. Finally, the $AVG_{codes}$ were merged, then concatenated to the samples in $\mathbf{D}_T$, and finally, such new feature space was used to perform the learning task in $\mathbf{D}_T$:

$$d(\mathbf{h_S}, \mathbf{h_T}) = \sqrt{\sum_{i=1}^{n}(h_{Si} - h_{Ti})^2}. \tag{8}$$

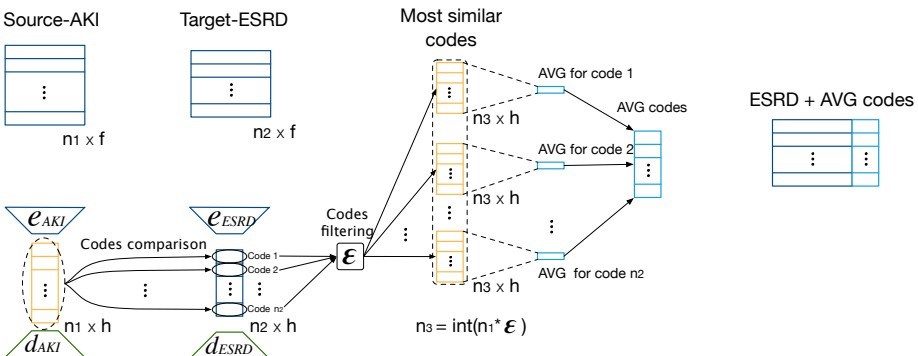

**Figure 6.** Scheme of proposed method for transferring $AVG_{codes}$ from $\mathbf{D}_S$ to $\mathbf{D}_T$ using TLAV.

### 2.5. Experimental Setup

This section presents all the necessary components and evaluates the proposed TL framework in predictive mortality models for patients in ESRD. Initially, we describe the datasets where the TL mechanisms are evaluated. Then, to compare the benefits of TLCO and TLAV, we modify HHTL for sample and feature space augmentation, as its initial learning task was to label unlabelled data in the target domain. In the description of the experiments it is explained how HHTL is modified. The experiments performed and their respective evaluations are presented at the end of the section.

#### Datasets

For this work, two datasets related to kidney disease were used. The learning tasks in both datasets are related to mortality prediction, one for the follow-up of patients in ESRD and the other one to patients with AKI in ICU. The objective is to improve predictive mortality models for patients in ESRD with data from patients with AKI. Next, they are described in detail.

**ESRD:** Information for $\mathbf{D}_T$ is part of a previous study for predicting mortality in ESRD patients [17]. Data were collected from the Information System of the Parc Tauli University Hospital, from the Haemodialysis (HD) Unit at the Nephrology Department from 2007 to 2018. Data transfer process was passed through the ethics committee (Code 2018/508) and subsequently anonymized following the usual protocol. Information from the follow-up of 261 patients in ESRD from the beginning of hemodialysis treatment until the deceased event was collected. The feature space includes categorical and continuous measurements from laboratory tests, diagnoses, and variables measured during the hemodialysis sessions. In total, there are 53 features. During their follow-up, such patients have generated 8229 samples. Four datasets were generated based on the date of death of the patients. Hence, the mortality models have labels associated with 1, 2, 3, and 6 months before the death event.

**AKI:** The dataset for $\mathbf{D}_S$ has been extracted from MIMIC-III database [18]. Such a massive database contains information from more than 40,000 patients in ICU. From MIMIC-III, patients with AKI were filtered based on the kidney disease improving global outcomes (KDIGO) clinical practice guideline [23]. Information from 4152 patients with 31 features were extracted. The total of samples in such cohort contains more than 125,000 samples.

Their follow-up includes demographics, diagnoses, laboratory tests, physiological measurements during the ICU stay, and the in-hospital mortality label.

## 3. Results

To evaluate the predictive capacity of the TL mechanisms in the mortality models for patients in ESRD, several experiments are defined based on the method of transferring knowledge. As AEs are the backbone of the proposed TL framework, we initially compared the performance of a deep AE with an mSDA applying TLCO and TLVA. Then, we compare the methods with HHTL. HHTL is modified in this work for sample and feature space augmentation. Next, the setups for mSDA and HHTL are listed:

- Deep AE vs. mSDA: We designed a baseline to choose which type of AE better suits the data. We train deep AEs with two hidden layers for the encoder and decoder functions. Then a two-level mSDA is trained. The codes are extracted from the deep AE to perform TLCO and TLAV. For mSDA, the hidden representations from the second level are extracted as codes, and TLCO and TLAV are applied to them.
- HHTL: HHTL has been widely compared with other approaches in the TL literature, showing a better performance than its competitors [22]. The modified versions of HHTL, for sample and feature space augmentation are based on the management of the hidden representations for the levels of the trained mSDAs. As stated in Section 2.2, such hidden representations are extracted from hidden layers to create new feature spaces:

$$\mathbf{Z}_T = \left[\mathbf{H}_{T,1}^\top \ldots \mathbf{H}_{T,K}^\top\right]^\top, \text{and } \mathbf{Z}_{S\to T} = \left[\left(\mathbf{H}_{S\to T,1}^*\right)^\top \ldots \left(\mathbf{H}_{S\to T,k}^*\right)^\top\right]^\top, \tag{9}$$

then sample augmentation is carried out adding samples from AKI to ESRD in their respective new feature space, i.e., $\mathbf{Z}_{samples} = [\mathbf{Z}_T; \ \mathbf{Z}_{S\to T}]$. As HHTL is a method to transfer samples, for feature space augmentation, as in the proposed approach, we use the averages of the most similar hidden representations from $\mathbf{Z}_{S\to T}$ to augment $\mathbf{Z}_T$, i.e., $\mathbf{Z}_{features} = \left[\mathbf{Z}_T \ \mathbf{AVG}_{\mathbf{Z}_{S\to T}}\right]$.

The performance of the experiments is evaluated on the learning task in ESRD. The area under the receiver operating characteristic (AUROC) curve is used as a metric to find the best models in the experiments. AUROC relates the sensitivity and specificity of a classifier. Its values lie between 0 and 1, with 1 being the perfect classifier and 0.5 being a random one. The baseline performance and classifiers used in this work are based on long short-term memory ANNs, used in the previous work for every mortality horizon [17]. All the reported experiments used five folds for cross-validation. Two sets of experiments have been defined to determine the performance of the proposed methods.

### 3.1. TLCO—Sample Augmentation

In ESRD data, the class imbalance varies according to the mortality horizon. Information on how the sample labels are computed can be found in the previous study [17]. Table 1 shows the class imbalance caused by each mortality horizon. To implement TLCO, initially, AEs with two hidden layers are trained for both datasets. Then, their codes are extracted. The hyperbolic tangent (Tanh) activation function is used for the hidden layers and the Sigmoid for the output layer in AKI. For the ESRD dataset, rectified linear unit (ReLU) activation function for hidden layers and Sigmoid at the output layer were used. Dropout of 0.1 and batch normalization were applied in the hidden layers of the AEs to avoid overfitting. Once the AEs are trained, the mapping matrix **G** is generated using codes from both domains. Then, the codes from AKI are transferred to the latent space of the ESRD domain using **G**. Finally, transformation is reconstructed using the decoding function of the trained AE in ESRD. For mSDA, a Tanh was used as a non-linear function to compute the codes. Next, three experiments are listed to find the best performance for the mortality predictors.

**Table 1.** Imbalance of samples for the prediction of mortality in patients in ESRD. **Class 0** and **Class 1** refer to samples in alive and deceased classes, respectively.

| Mortality | Class 0 | Class 1 | Imbalance (%) |
|:---:|:---:|:---:|:---:|
| 1 | 7734 | 495 | 93.6 |
| 2 | 7488 | 741 | 90.1 |
| 3 | 7251 | 978 | 86.5 |
| 6 | 6632 | 1597 | 75.9 |

- Code dimension: The dimensions of codes in both domains are evaluated to find a high-level representation of the data that allows us to transfer valuable information. Thus, the combination of dimensions that presents the best overall performance for the prediction task is empirically found. In Figure 7a, it is denoted the dimension of the codes for the deep AE in AKI and ESRD as $S\_*$ and $T\_*$, where $*$ refers to the dimensions of the code, e.g., $S\_30$ and $T\_40$ refers to the combination of having trained AEs with codes of dimension 30 and 40 for AKI and ESRD, respectively. It is also shown the performance of mSDA. Moreover, it should be noted that in mSDA, the dimension of the codes has the same input data dimension, which is why only one predictor is observed for mSDA in the figure. In addition, it can be appreciated that most of the combinations present a higher performance than the baseline one. Although mSDA outperforms better than most predictors, the deep AE with 30 and 80 codes in AKI and ESRD offers a better predictive capacity than mSDA.

- Sample augmentation in ESRD: This experiment evaluates how the increase in samples in the training set affects the predictive models of mortality in ESRD. For this experiment, three possible scenarios were defined. Initially, the data imbalance in ESRD is intentionally increased. Thus, only Class 0 in AKI samples are transferred to the ESRD training set. This transfer is carried out to evaluate whether an adverse effect is linked to the increase in data imbalance. In the second scenario, the training set samples are increased, but only those that belong to AKI Class 1 are transferred. In this case, the aim is to balance the imbalanced class. Finally, in a third scenario, both classes are transferred from AKI to ESRD. Therefore, we evaluate both the effect of the increase in samples and the reduction in the data imbalance in the predictive models. Table 2 shows how the data imbalance varies for each scenario. In Figure 7b, it can be appreciated that increasing samples in the training set of the ESRD data does not imply, in most of the scenarios, a reduction in the predictive models performance. On the other hand, when the number of samples increases, the learning models present a better predictive capacity considering the imbalance ratio.

- Comparing with HHTL: To evaluate the performance of HHTL, the number of transferred samples was adjusted following the third scenario in the previous experiment. Thus, in Figure 7c it can be appreciated that although HHTL for upsampling or HHTL4S improves the base predictive models, it has a lower performance than that found by deep AE.

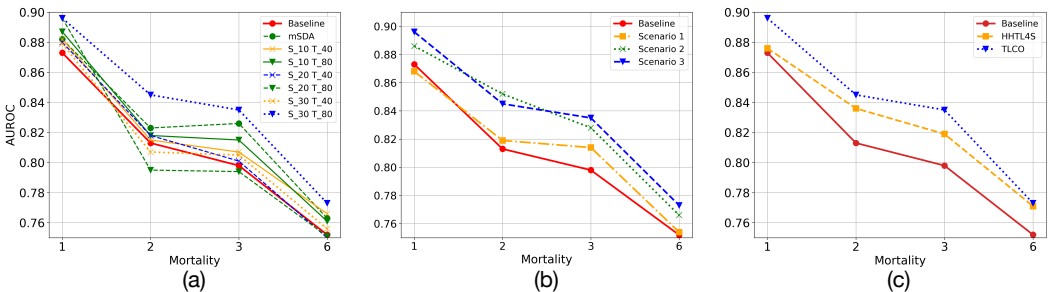

**Figure 7.** Comparison results transferring samples from AKI to ESRD using TLCO: (**a**) shows the performance of mSDA and TLCO modifying dimension of codes in source (S) and target (T) domains; (**b**) shows three possible scenarios to tackle data imbalance in ESRD; (**c**) compares proposed solution with HHTL.

**Table 2.** Imbalance in ESRD generated by increasing training samples in ESRD from AKI. Scenarios 1, 2, and 3 refer to the transfer of samples from Classes 0, 1, and combining both classes, respectively.

| Mortality | Generated Data Imbalance (%) | | |
|:---:|:---:|:---:|:---:|
| | Scenario 1 | Scenario 2 | Scenario 3 |
| 1 | 95.2 | 73.4 | 80.0 |
| 2 | 92.6 | 69.2 | 77.1 |
| 3 | 90.1 | 74.9 | 74.2 |
| 6 | 82.7 | 52.3 | 65.7 |

### *3.2. TLAV—Feature Space Augmentation*

In the performance evaluation of TLAV, the same hyperparameters for training AEs as in TLCO are used for TLAV and its competitors. With TLAV, the augmented feature space is based on the computation of $AVG_{codes}$. As a recall, such $AVG_{codes}$ are computed based on the comparison of each code in ESRD with all the codes in AKI. Each comparison generates a set of codes that are filtered by a similarity threshold ($\epsilon$) and summarized into $AVG_{codes}$. Parametric analysis and comparison with mSDA configuration and HHTL for feature augmentation (HHTL4F) are carried out. Thus, three experiments were performed to find the best models that enhance the learning task in ESRD. Next, they are addressed.

- Code dimension: The first parameter that controls the behavior of TLAV is the dimension of the codes ($dim\_h$). This parameter reflects the ability of AEs to represent information in latent spaces under the TL methodology of TLAV. In this experiment, $\epsilon$ is set to 0.4. Figure 8 shows scenarios where the input information is compressed or dispersed according to the value of $dim\_h$. It can be appreciated that bottleneck type deep AEs offer better overall performance than sparse type deep AEs. The best solution is the one with $dim\_h = 10$.

- Tuning similarity threshold ($\epsilon$): With Euclidean distances from ESRD and AKI codes, a proportion of these codes is chosen using $\epsilon$. $\epsilon$ controls the amount of more similar AKI codes used to compute the average one. Once every set of codes from AKI are extracted, their $AVG_{codes}$ are computed and used to increase the feature space for each ESRD sample. Table 3 shows the performance of the predictive models varying $\epsilon$. It can be appreciated that increasing the number of codes for the computation of their average reflects a slight improvement in the predictive models. However, from an $\epsilon$ of 0.3 or 0.4, more codes do not imply a considerable increase in the predictive models. Compared with its competitors, TLAV based on deep AEs presents a better performance when more codes are included for the average computation. Using the three methods, taking 40% of the most similar AKI codes for each ESRD code presents the most balanced performance for mortality prediction. TLAV with deep AEs is the best option to increase the feature space in ESRD.

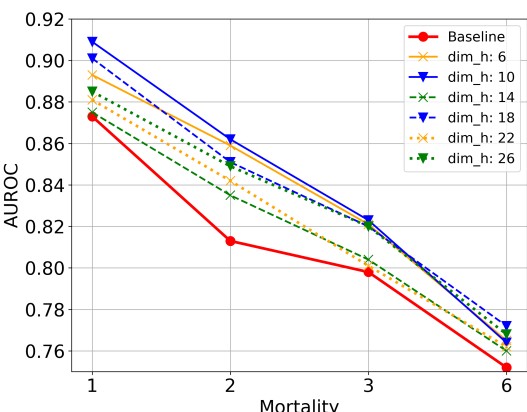

**Figure 8.** Evaluation of TLAV changing the dimension of the codes.

**Table 3.** Comparison results applying TLAV with mSDA and HHTL4F using $AVG_{codes}$ concept and varying similarity threshold $\epsilon$. In bold, the best predictive models for each mortality horizon.

| $\epsilon$ | TL Method | Mortality | | | |
| | | 1 | 2 | 3 | 6 |
|---|---|---|---|---|---|
| 0.01 | mSDA | 0.857 | 0.839 | 0.816 | 0.761 |
| | HHTL4F | 0.878 | 0.824 | 0.820 | 0.757 |
| | TLAV | 0.887 | 0.849 | 0.816 | 0.758 |
| 0.1 | mSDA | 0.856 | 0.840 | 0.809 | 0.761 |
| | HHTL4F | 0.879 | 0.831 | 0.818 | 0.759 |
| | TLAV | 0.891 | 0.854 | 0.816 | 0.763 |
| 0.2 | mSDA | 0.859 | 0.834 | 0.811 | 0.760 |
| | HHTL4F | 0.891 | 0.834 | 0.819 | 0.758 |
| | TLAV | 0.901 | 0.857 | 0.820 | 0.761 |
| 0.3 | mSDA | 0.863 | 0.841 | 0.820 | 0.760 |
| | HHTL4F | 0.895 | 0.837 | 0.822 | 0.758 |
| | TLAV | 0.906 | 0.860 | **0.823** | **0.765** |
| 0.4 | mSDA | 0.877 | 0.842 | 0.819 | 0.758 |
| | HHTL4F | 0.894 | 0.835 | 0.821 | 0.760 |
| | TLAV | **0.909** | **0.862** | **0.823** | 0.763 |
| 0.5 | mSDA | 0.875 | 0.842 | 0.818 | 0.759 |
| | HHTL4F | 0.891 | 0.836 | 0.819 | 0.759 |
| | TLAV | 0.904 | 0.861 | 0.821 | 0.765 |

### 3.3. TLAV—HHTLM

In the last experiment, TLAV is combined with TLCO. Such a combination is performed in a cascade way. The parameters that control TLCO and TLVA are found in previous experiments. Thus, in the first stage, the ESRD feature space increases using TLAV. Then, TLCO is applied to this new version of ESRD to increase the number of samples. Table 4 presents the performance of the combination, compared to the literature methods and the best predictors by TLAV and TLCO separately. It can be seen that the combination of the two proposed methods has a considerable influence on the performance of the predictive models for short-term mortality.

**Table 4.** Final comparison of the proposed TL framework. TLAV-TLCO is the cascade version of the TL proposed methods. In bold the most relevant results for each mortality predictor.

| Mortality | Baseline | TLCO | TLAV | TLAV-TLCO |
|:---:|:---:|:---:|:---:|:---:|
| 1 | 0.873 | 0.891 | 0.909 | **0.939** |
| 2 | 0.813 | 0.845 | 0.862 | **0.909** |
| 3 | 0.798 | 0.838 | 0.823 | **0.853** |
| 6 | 0.752 | **0.778** | 0.765 | 0.764 |

## 4. Discussion

This work has explored a novel TL alternative based on sample and feature space augmentation based on TLCO and TLVA. Information was transferred from a massive data source and improved predictive mortality models in ESRD patients. The transferred information was extracted in the codes of both domains. It was shown that transferring knowledge from another data source directly improves the learning models using codes from AEs. The conducted experiments have shown that deep AEs extract better complex relationships for the available domains than mSDAs.

For the experiments related to sample augmentation, it was found that TLCO provided an improvement from 2–5% in AUROC when both classes are transferred from AKI. It was evidenced that increasing just the imbalance in most models does not deteriorate the predictions' performance. Reducing data imbalance provides a considerable improvement for the learning models, although the predictive ability in the data increases considerably when both classes are included in the upsampling.

For the case of feature space augmentation, it was evidenced that increasing the information in ESRD with the $AVG_{codes}$ improves the performance of the learning models even when other alternatives such as mSDA or HHTL are used. Moreover, TLAV was shown to generalize better than TLCO in predictive models for a 2-month mortality horizon. It was evidenced that the dominant parameter that controlled the performance of the learning models was the dimension of the codes. In the case of threshold $\epsilon$, from the inclusion of 40% of the codes from AKI, it is enough to guarantee an increase in performance among 2–6% compared to the baseline models.

Finally, the results obtained showed that the proposed framework can improve the predictive capacity of mortality models in ESRD and that they can be complementary to each other. If these two are combined, the performance of these models increases considerably (6–11%). Such improvements in the performance of the mortality predictors could imply that incorporating this type of solution into clinical setting brings us closer to incorporating data-driven solutions to support medical staff in the early detection of events such as mortality.

**Author Contributions:** Conceptualization, E.M. and A.M.; data curation, E.M., J.I. and A.M.; formal analysis, A.M.; investigation, E.M., J.S. and J.L.V.; methodology, E.M., J.S., J.L.V., J.I. and A.M.; visualization, A.M.; writing—original draft, E.M.; writing—review and editing, E.M., J.S. and J.L.V. All authors have read and agreed to the published version of the manuscript.

**Funding:** This work is supported by the Spanish Government under Project TEC2017-84321-C4-4-R co-funded with European Union ERDF funds and also by the Catalan Government under Project 2017 SGR 1670.

**Institutional Review Board Statement:** The study was conducted in accordance with the Declaration of Helsinki, and approved by the Ethics Committee of Parc Taulí Hospital Universitari, Institut de Investigació i Innovació Parc Taulí I3PT (protocol code 2018/508).

**Informed Consent Statement:** Informed consent was obtained from all subjects involved in the study.

**Data Availability Statement:** AKI data presented in this study are openly available in MIMIC-III at https://doi.org/10.13026/C2XW26. ESRD data presented in this study are available on request from the corresponding author. The data are not publicly available due to privacy.

**Conflicts of Interest:** The authors declare no conflict of interest.

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
