# Peer review of "Transfer Learning Improving Predictive Mortality Models for Patients in End-Stage Renal Disease"

_electronics, doi:10.3390/electronics11091447_

Round 1

Reviewer 1 Report

The authors have proposed the transfer learning model based on sample and feature space augmentation. They have used latent space mapping matrix to transfer samples. The computation of average of most similar codes from one domain forms the basis of feature space augmentation in target domains. the proposed framework has been evaluated to predict mortality in patients in end stage renal disease transferring information related to mortality of patients with acute kidney injury from MIMIC-III database.authors claim 6-11% improvement in the performance. It is a good piece of research work on transfer learning and its application. The authors should add some limitations of their proposed model and future directions in their discussion/conclusion part of their research paper.

Author Response

Dear reviewer,

First of all, the authors of the manuscript would like to thank the reviewers and editors for their comments and suggestions. We very much appreciate the time you have spent as your suggestions were found to be very useful to improve the quality of the manuscript.

In the attached document we address your comments.

Best regards,

Edwar Macias 

Reviewer 2 Report

The manuscript entitled “Transfer learning improving predictive mortality models for patients in end-stage renal disease” has been investigated in detail. The topic addressed in the manuscript is potentially interesting and the manuscript contains some practical meanings, however, there are some issues which should be addressed by the authors:

  • The "Abstract" section can be made much more impressive by highlighting your contributions. The contribution of the study should be explained simply and clearly.
  • The readability and presentation of the study should be further improved. The paper suffers from language problems.
  • The “Introduction” section needs a major revision in terms of providing more accurate and informative literature review and the pros and cons of the available approaches and how the proposed method is different comparatively. Also, the motivation and contribution should be stated more clearly.
  • The importance of the design carried out in this manuscript can be explained better than other important studies published in this field. I recommend the authors to review other recently developed works.
  • The performance of the proposed method should be better analyzed, commented and visualized in the experimental section.
  • What makes the proposed method suitable for this unique task? What new development to the proposed method have the authors added (compared to the existing approaches)? These points should be clarified.
  • “Discussion” section should be added in a more highlighting, argumentative way. The author should analysis the reason why the tested results is achieved.
  • The authors should clearly emphasize the contribution of the study. Please note that the up-to-date of references will contribute to the up-to-date of your manuscript. The studies named- Detection of solder paste defects with an optimization‐based deep learning model using image processing techniques; Optimization of deep learning model parameters in classification of solder paste defects- can be used to explain the proposed method in the study or to indicate the contribution in the “Introduction” section.
  • How to set the parameters of proposed method for better performance?
  • It will be helpful to the readers if some discussions about insight of the main results are added as Remarks.

This study may be proposed for publication if it is addressed in the specified problems.

Author Response

Dear Reviewer,

First of all, the authors of the manuscript would like to thank you for your comments and suggestions. We very much appreciate the time you have spent as your suggestions were found to be very useful to improve the quality of the manuscript.

In the attached document we have addressed your comments.

Best regards,

Edwar Macias

Round 2

Reviewer 2 Report

It is acceptable in the present form.